# Rapid mechanosensitive migration and dispersal of newly divided mesenchymal cells aid their recruitment into dermal condensates

Jon Riddell[1], Shahzeb Raja Noureen[2], Luigi Sedda[3], James D. Glover[1], William K. W. Ho[1], Connor A. Bain[4], Arianna Berbeglia[1], Helen Brown[1], Calum Anderson[4], Yuhang Chen[4], Michael L. Crichton[4], Christian A. Yates[2], Richard L. Mort[5]*, Denis J. Headon[1]*

1 The Roslin Institute and Royal (Dick) School of Veterinary Studies, University of Edinburgh, Edinburgh, United Kingdom, 2 Department of Mathematical Sciences, University of Bath, Claverton Down, Bath, United Kingdom, 3 Lancaster Ecology and Epidemiology Group, Lancaster Medical School, Lancaster University, Lancaster, United Kingdom, 4 School of Engineering and Physical Sciences, Heriot-Watt University, Edinburgh, United Kingdom, 5 Division of Biomedical and Life Sciences, Faculty of Health and Medicine, Lancaster University, Lancaster, United Kingdom

* r.mort@lancaster.ac.uk (RLM); denis.headon@roslin.ed.ac.uk (DJH)

**Data Availability Statement:** The authors confirm that all data underlying the findings are fully available without restriction. All relevant data are

## Abstract

Embryonic mesenchymal cells are dispersed within an extracellular matrix but can coalesce to form condensates with key developmental roles. Cells within condensates undergo fate and morphological changes and induce cell fate changes in nearby epithelia to produce structures including hair follicles, feathers, or intestinal villi. Here, by imaging mouse and chicken embryonic skin, we find that mesenchymal cells undergo much of their dispersal in early interphase, in a stereotyped process of displacement driven by 3 hours of rapid and persistent migration followed by a long period of low motility. The cell division plane and the elevated migration speed and persistence of newly born mesenchymal cells are mechano-sensitive, aligning with tissue tension, and are reliant on active WNT secretion. This behaviour disperses mesenchymal cells and allows daughters of recent divisions to travel long distances to enter dermal condensates, demonstrating an unanticipated effect of cell cycle subphase on core mesenchymal behaviour.

## Introduction

Many vertebrate organs, including lung, intestine, skin, kidney, and mammary gland, are composites of an epithelium and a connective tissue stroma. The stromal component originates from embryonic mesenchyme, consisting of motile cells dispersed within an extracellular matrix. In the skin, the mesenchyme is largely populated with embryonic fibroblasts, while also carrying adipocytes, blood vessels, lymphatics, nerves, macrophages, and melanocytes [1]. On the trunk, the dermal mesenchyme of the back originates from the somites [2], with surface ectodermal WNT signals driving specification of the dermal fibroblast precursors [3]. In the skin, the dermal mesenchymal cells acquire different fates based on their depth relative to the epithelium, with WNT signals maintaining the papillary dermis fate in the upper dermis and

within the paper and its Supporting Information files. Source code has been deposited in Zenodo repositories https://doi.org/10.5281/zenodo.8229691 and https://doi.org/10.5281/zenodo.8229341.

**Funding:** This work was supported by BBSRC awards BB/T007788/1 and BBS/E/D/10002071 received by DJH, and by North West Cancer Research Fund award CR1132, Royal Society award RGS\R2\212427 and Dr. Philip Welch STEM research project funding received by RLM. SRN is supported by a scholarship from the EPSRC Centre for Doctoral Training in Statistical Applied Mathematics at Bath (SAMBa), under the project EP/S022945/1. This research made use of the Balena High Performance Computing (HPC) Service at the University of Bath. CAB is supported by a Carnegie Trust for Scotland PhD studentship. The funder had no role in study design, data collection and analysis, decision to publish, or preparation of the manuscript.

**Competing interests:** The authors have declared that no competing interests exist.

**Abbreviations:** A-P, anterior-posterior; AWERB, the Animal Welfare and Ethical Review Body; DMEM, Dulbecco's Modified Eagle Medium; FBS, foetal bovine serum; MSD, mean squared displacement; NARF, National Avian Research Facility.

suppressing the acquisition of adipocyte identity, resulting in dermal adipocytes forming only in deeper dermis [4,5]. In adult skin, dermal fibroblasts decrease in number with age and are nonmotile, while embryonic populations undergo extensive movement [6,7].

Cells in the upper dermis form condensates at sites of emerging skin appendages, such as hair and feather follicles, morphologically similar to the condensates that form the mesenchymal component of intestinal villi, tooth buds, and cartilaginous template of the skeleton [8–10]. These dermal condensates are formed through recruitment of mesenchymal cells by focally produced epithelial signals, notably FGF20 and SHH [3,7,11–13], though autonomous coalescence of mesenchymal cells can be triggered under certain conditions without epithelial signals [7,13]. Once formed, dermal condensates induce fate commitment and appendage growth in their overlying epithelium [14–17]. In embryonic skin, proliferation of mesenchymal cells occurs widely outside condensates [12,18] but rarely once cells enter them [19,20]. A select, highly proliferative, fibroblast population in the peri-condensate zone has been hypothesised to provide cells for condensate construction, based on cell labelling with thymidine analogues and gene expression analyses [19,21].

Here, we delineate embryonic mesenchymal cell behaviour by direct observation in developing skin, finding a signature WNT-dependent mechanosensitive migratory mechanism that both disperses mesenchymal cells in the general dermis but also aids their recruitment into developing condensates.

## Results

### Rapid displacement of newly born mesenchymal cells in embryonic skin

To define individual cell behaviours during mesenchymal development, we analysed time-lapse confocal imaging of ex vivo cultured TCF/Lef::H2B-GFP mouse skin containing H2B-GFP labelled mesenchymal cell nuclei [22], allowing us to track their movement throughout the cell cycle. Our observations first indicated that movement and mitosis of dermal cells occur in a planar manner in parallel to the epithelium at all depths of mesenchyme (Figs 1A, S1A, and S1B and S1 Video).

We assessed the behaviour of dividing cells and of cells that were not observed to divide within the duration of observation (hereafter referred to as non-dividing cells) in our time-lapse movies. Cells that did not divide over the 25 h duration of the time-lapse migrated without preferred direction (Fig 1B). Occasional cells (<1/1,000) migrated at high speed throughout the imaging period (S2 Video), but most moved at a mean speed of 0.02 μm min⁻¹ (Fig 1C). Upon and following mitosis, we observed a stereotyped daughter cell displacement, characterised by rapid separation at cytokinesis, often followed by a short pause and then a 180 min period of rapid, persistent migration in early interphase, before returning to the behavioural characteristics of non-dividing cells (Figs 1C–1G and S1C and S3 Video). As a population, the diffusion coefficient of daughter cells in this window was significantly greater, resulting in a mean Euclidean displacement approximately 50% greater than in the equivalent non-dividing cells over the same time period (Fig 1D–1G). To determine if this newly born cell behaviour was conserved in vertebrates, we analysed dividing mesenchymal cell behaviour in chicken skin using recombinant cell permeable TAT-Cre–induced Chameleon chicken cells [12] (S2A Fig and S4 Video) in explant skin culture. We observed a similar increase in speed, Euclidean displacement, and diffusion coefficient of newly born cells, though we did not detect an increased persistence of movement (S2B–S2G Fig). In line with these short-term observations, longer term lineage tracing of cells derived from the somites or the surface ectoderm using TAT-Cre induction in the Chameleon chicken line over 4 days revealed the formation of elongated contiguous keratinocyte clones in the epidermis running dorsolaterally, while

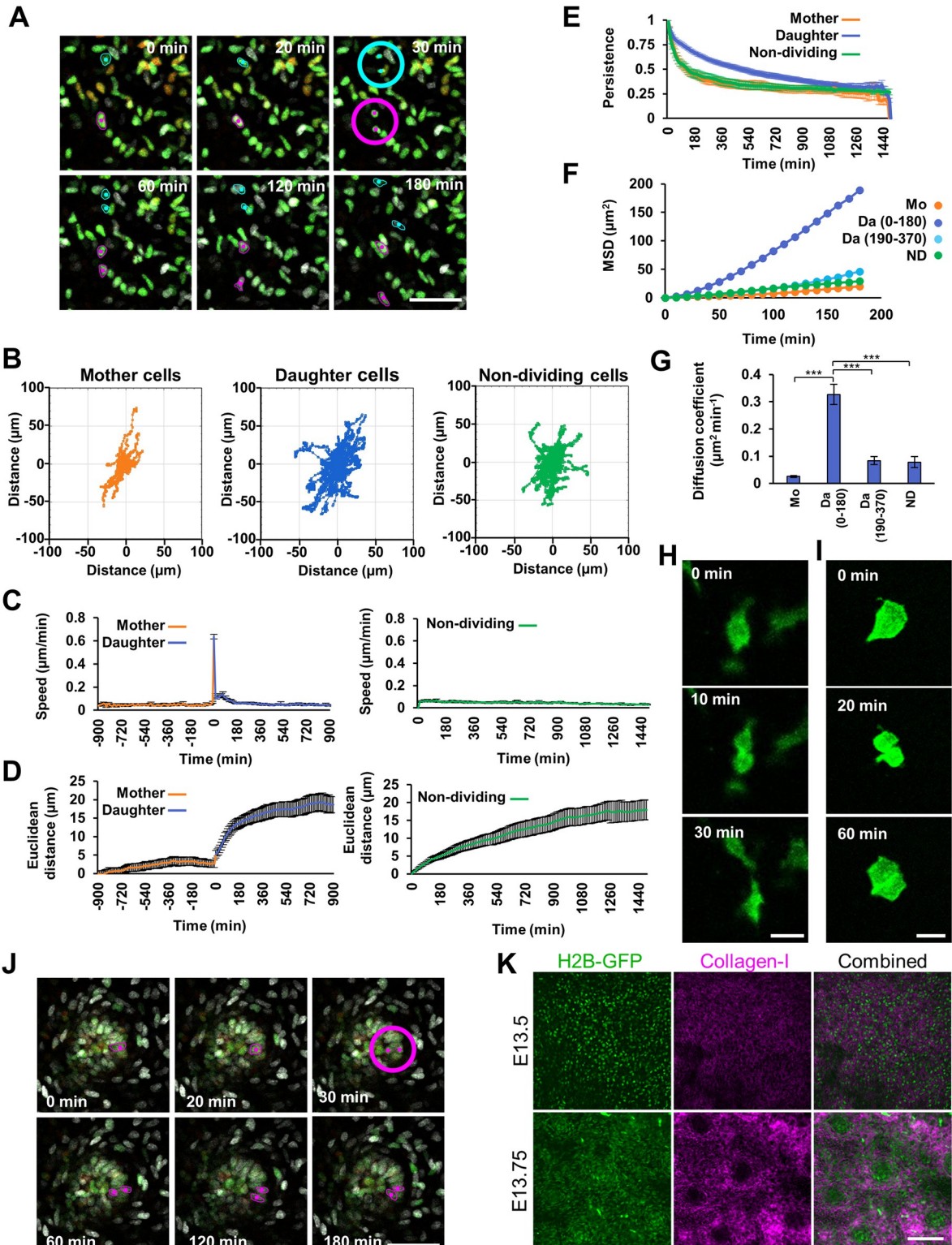

**Fig 1. Mesenchymal mitosis is followed by rapid daughter cell displacement. (A)** Time-lapse series of embryonic day (E) 13.5 TCF/Lef::
H2B-GFP mouse skin explant culture. Magenta and cyan dots highlight nuclei undergoing mitosis, circles indicate the point at which cells
divide, z-planes are colour-coded (red = closest to epithelium, green = middle, grey = deepest). **(B)** Diffusion plots showing the migration
direction and dispersion of tracked mother (left; *n* = 45), daughter (middle; *n* = 89), and non-dividing (right; *n* = 51) cells from a single
skin explant. **(C, D)** Plots showing speed **(C)** and Euclidean distance travelled **(D)** by tracked dividing (left panels; point of mitosis set at

time = 0) and non-dividing (right panels) cells. **(E)** Persistence (Euclidean/accumulated distance) of mother, daughter, and non-dividing cells against time. **(F)** MSD of mother cells (Mo; $n = 45$) for 180 min prior to division, daughter cells for 180 min after division, daughter cells for 180–360 min after division (Da; $n = 89$ in both windows), and non-dividing cells (ND; $n = 51$) for a single representative skin explant. **(G)** Mean diffusion coefficient (the slope of the line in **F**) of mother cells (Mo), daughter cells (Da) for 180 min after division, daughter cells for 180–360 min after division, and non-dividing cells (ND). A one-way analysis of variance (ANOVA ($p < 0.001$)) followed by pairwise post hoc Tukey's honestly significant difference tests revealed a significant difference between newly born daughters (0–180 min) and all other groups (***$p < 0.001$). **(H, I)** Single planes from confocal time-lapse series of a cultured E13.5 mTmG mouse skin showing **(H)** a dividing mesenchymal cell and resulting daughter cells and **(I)** a dividing peridermal cell. **(J)** Time series of mitosis in a dermal condensate. Magenta dots highlight mother and daughter nuclei. z-planes colour-coded as in **A**. **(K)** Single planes from confocal imaging of Collagen-I immunofluorescence in E13.5 and E13.75 TCF/Lef::H2B-GFP dermal mesenchyme. **C**, **D**, **E**, and **G** time-lapse videos $n = 8$ from 4 independent embryos, mean number of dividing cells tracked per video = 42, non-dividing cells tracked per video = 50. The raw tracking data for **B**–**G** can be found in S1 Data. Error bars represent the SEM. Scale bar in **A** = 50 μm; scale bar in **H** and **I** = 20 μm; scale bar in **J** = 50 μm; scale bar in **K** = 100 μm. MSD, mean squared displacement; SEM, standard error of the mean.

mesenchymal clones, although elongated and running dorsolaterally, were not contiguous (S2H Fig), consistent with dispersal of mesenchymal cells immediately upon mitosis. Similarly, when we transplanted a somite from a tdTomato (TPZ) chicken into a GFP chicken, we observed a dorsolateral streak of tissue expansion originating from the somite (S2I Fig), reflecting a favoured net direction of cell displacement in the growing embryo.

To assess individual cell shape through mitosis, we induced sparse labelling of membrane localised MARCKS-EGFP in mesenchymal, peridermal, and basal epidermal cells using the *R26R-mTmG* mouse line [23]. We induced labelling using TAT-Cre in E13.5 skin, then cultured and confocal imaged for 16 h. We noted that dermal mesenchymal cells typically have planar cell projections, running parallel to the epithelium. These projections are often, but not always, lost upon cell division, with rounding of the cells typically occurring as they enter mitosis (Fig 1H). However, several instances of retention of major cellular processes, followed by migration along their tracks, were evident (S5 Video). The behaviour we observed in mesenchymal cells was distinct from that in epithelia, where we observed divisions of periderm and of basal epidermal cells which produced adjacent daughter cells (Fig 1I and S6 Video), in agreement with the contiguous epithelial clones we observed in chicken embryos over a longer timescale (S2H Fig).

Mesenchymal fibroblasts contribute to hair follicles by forming dermal condensates, precursors to the dermal papilla [24]. Though the embryonic dermal mesenchyme is highly proliferative, cells in the dermal condensates are long recognised as being largely quiescent [11,19,21]. However, in our imaging experiments, we observed some mitoses taking place within dermal condensates (7 division events observed from 8 condensates over 25 h of imaging; Fig 1J and S7 Video). In sharp distinction to the general dermis, these mitoses resulted in very limited cell displacement, producing a pair of daughter cells that remained in close proximity to one another, suggesting that the local extracellular environment influences the behaviour of newly born cells in the mesenchyme. To assess this, we immunostained embryonic mouse skin and found that Collagen-I was absent from the dermal condensates, revealing their distinct extracellular matrix environment (Figs 1K and S3).

## Rapid movement of the newly born mesenchymal cells increases dermal condensate entry

Dermal mesenchymal cells are drawn to condensates by soluble signals, notably FGF20 and SHH, from the overlying epithelial placodes [7,11]. In agreement with previous findings [19,21], our imaging experiments revealed that cells within the first 180 min post-division were more likely to enter a dermal condensate than non-dividing cells, with the rate of entry declining to that of the general interphase population thereafter (Fig 2A and S1 Table).

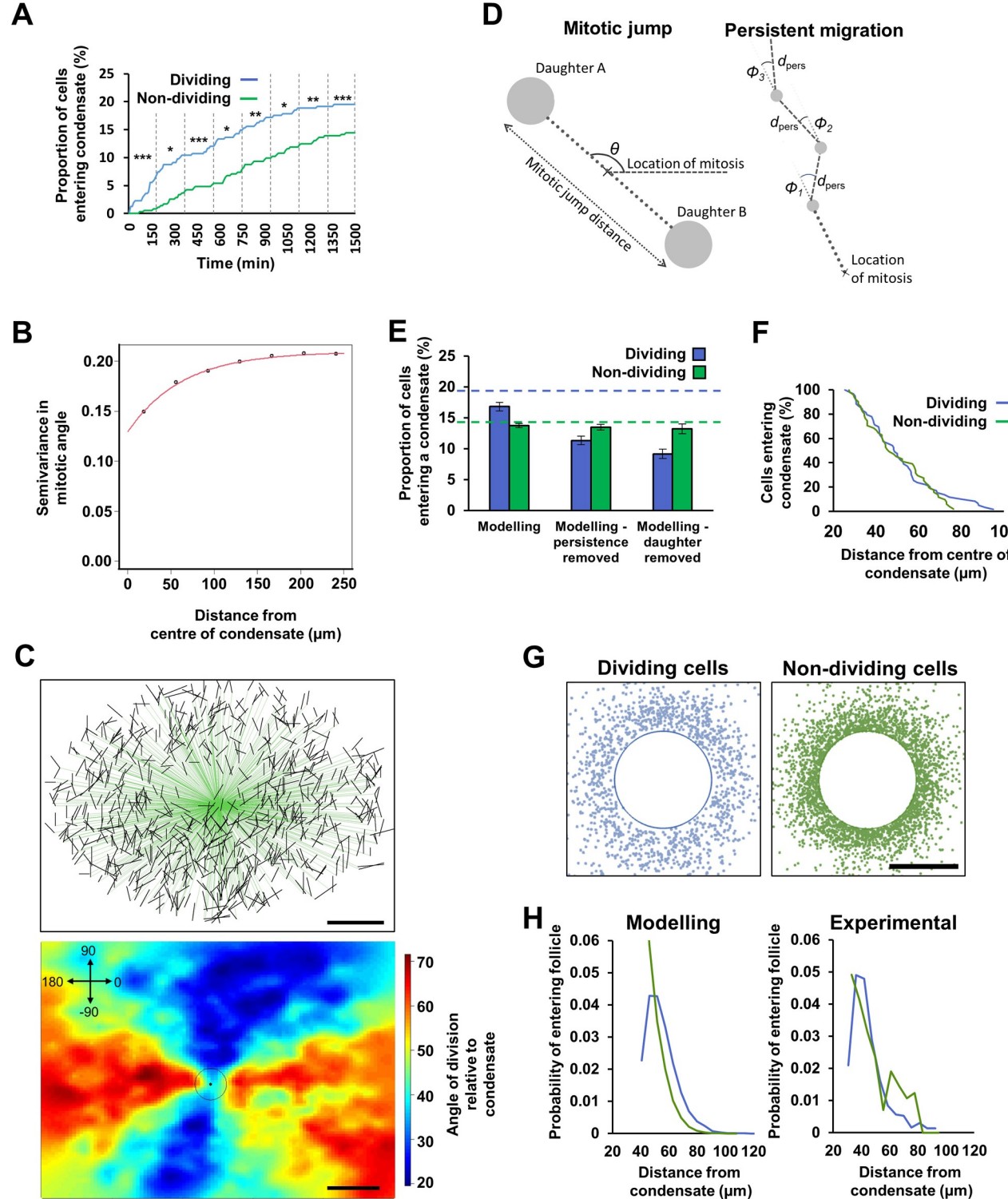

**Fig 2. Newly born mesenchymal cells have increased probability of dermal condensate entry.** (**A**) Timing and proportion of tracked dividing and non-dividing cells entering the condensates as a cumulative percentage. Vertical dashed lines and asterisks indicate time windows and significance levels for a Fisher's exact test (S1 Table). Time 0 is the point of mitosis for dividing cells. Cells from 0 to 180 min post-mitosis have an increased rate of condensate entry compared to all other cells (*$p < 0.05$, **$p < 0.01$, ***$p < 0.001$). Time-lapse videos $n = 8$ from 4 independent embryos, mean number of dividing cells tracked per video = 42, non-dividing cells tracked per video = 50. (**B**) Averaged spatially dependent variance (variogram) in mitosis angles. Approximately 70% of the variance in the mitosis angle occurs within a distance of 0 to 60 μm from the condensate (Monte Carlo probability $p < 0.001$). Time-lapse videos $n = 5$ from 5 independent embryos, mean number of mitosis angles plotted per video $n = 622$. (**C**) Upper:

Cell division angles relative to their nearest condensate for a single representative time-lapse sequence. Black lines indicate direction of mitosis, green lines connect division events to the condensate centre, from which angles relative to the condensate were calculated. Lower: heat map of the cell division angles relative to nearest dermal condensate for the same dataset (black circle = centre). The coordinate system used is indicated in the top left-hand corner. The raw mitosis angle and follicle position data for **B** and **C** can be found in S2 Data. **(D)** Schematic of the agent-based model of mesenchymal cell division composed of a mitotic jump (left panel) displacing the daughters in diametrically opposite directions (at angle $\theta$ from the horizontal) followed by a persistent random walk (right panel—dashed lines). The angle of migration of step i, relative to the direction of the mitotic jump, is represented by $\Phi_i$, and the distance travelled in each step is represented by $d_{pers}$. **(E)** Plot showing the percentage (+/− SEM; $n = 8$) of simulated dividing and non-dividing cells entering a condensate in the model. Dashed blue and green lines indicate the proportion of dividing and non-dividing cells entering condensates, respectively, from experimental data. **(F)** Plot showing percentage of dividing and non-dividing cells entering a condensate against their initial position relative to the condensate centre from experimental data. Lineages with recent divisions can be recruited from further away. Time-lapse videos $n = 8$ from 4 independent embryos, mean number of dividing cells tracked per video = 42, non-dividing cells tracked per video = 50. **(G)** Initial locations of dividing (left) and non-dividing (right) agents that ultimately enter a condensate, from simulation. **(H)** For cells entering follicles, the probability density of entry from a given starting distance for dividing (blue) and non-dividing (green) cells–modelled (left) and experimental (right). The raw tracking data for **A** and **E**–**H** can be found in S1 Data. Scale bars in **C** = 100 μm; scale bar in **G** = 50 μm. SEM, standard error of the mean.

We next examined whether the dermal condensate influences the orientation of nearby cell divisions. A permutation analysis of mitosis angles showed that, at distances closer than 177 μm, the distribution of mitosis angles cannot be explained by random process. We observed that 70% of the variability in mitosis direction was affected by the presence of a condensate up to a mean of 60 μm from its centre, and the remaining 30% up to a mean of 177 μm from its centre (Fig 2B). We also observed that mitoses occur at close to an orthogonal tangent to the condensate perimeter (Fig 2C and S8 Video). Therefore, dermal condensates influence the division angle of mesenchymal cells in their immediate proximity. However, dividing cells do not orientate directly towards the condensate, and thus, this effect does not explain the higher proportion of dividing cells ultimately entering the condensate (19.4%), compared to non-dividing cells (14.3%; Fig 2A).

In order to investigate the mechanisms underlying the observed recruitment bias in favour of newly born mesenchymal cells to condensates, we built an agent-based mathematical model (Fig 2D and S1 Appendix). Dermal fibroblast cells (the agents) were initialised uniformly at random on a square domain (representing the dermis) with periodic boundary conditions that contained preformed condensate recruitment sites with fully absorbing boundary conditions. Cells encountering the condensate were immediately absorbed and removed from the simulation, consistent with our findings that cells are not observed exiting established condensates. Cell movement on the domain was an off-lattice diffusive random walk, with migration parameters being those directly measured from the non-dividing mesenchymal cell population (Fig 1F). To model mitosis, cells were selected at random, then simulated to undergo a proliferation event forming 2 daughters, which then underwent mitotic displacement over 180 min according to our measured cell movement parameters (Fig 2D and S1 Appendix). In this model, newly born mesenchymal cells have a greater likelihood of condensate entry than non-dividing cells as a result of both cell number increase (through division) and the increased displacement of the resulting daughters, making them more likely to encounter a condensate (Fig 2E). Our modelling predicted that the probability of dividing cells being recruited from short distances would be lower than for non-dividing cells but that they could be recruited from a greater distance, as suggested by our cell tracking data (Fig 2F) and confirmed by plotting the probability density (Fig 2G and 2H).

## Mitotic displacement of mesenchymal cells is locally coordinated in intact embryos

In our initial imaging of skin cultures, we noted that mitotic angles tended to be aligned with one another across the field of view (Fig 3A). To determine whether this observation was

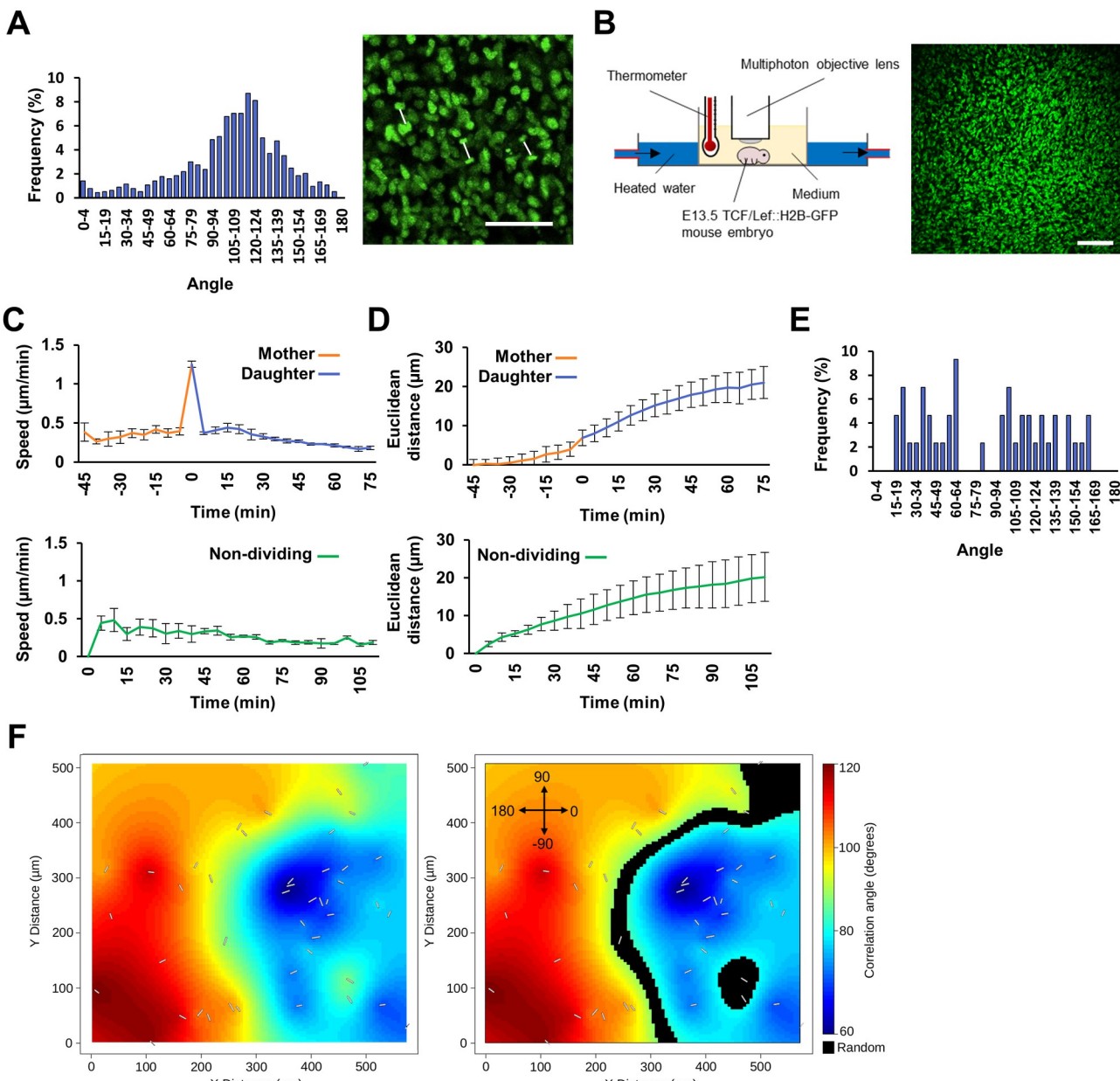

**Fig 3. Mitotic orientations are locally aligned in embryonic mesenchyme.** (**A**) Distribution of mitotic angles (*n* = 1,135) in a TCF/Lef::H2B-GFP skin explant culture (left panel) and representative frame showing coherence of mitotic orientations (right panel). Daughter nucleus pairs are connected by white lines, showing the angle of mitosis. The raw mitosis angle data for **A** can be found in S2 Data. (**B**) Schematic of whole embryo culture and multiphoton imaging with corresponding image of the trunk skin of an E13.5 TCF/Lef::H2B-GFP embryo. (**C, D**) Speed (**C**) and Euclidean distance travelled (**D**) by tracked dividing and non-dividing cells. Whole embryo time-lapse from 3 independent embryos, average number of divisions tracked/ video = 65, non-dividing cells tracked/video = 50. In dividing cell plots, time 0 = mitosis. (**E**) Distribution of angles (*n* = 43) of division from TCF/Lef:: H2B-GFP whole mouse embryo imaging; 0 and 180 degrees are parallel to the A-P axis. (**F**) Spatial distribution of mitotic angles in **E** (white overlaid lines), with heat map showing areas where angles are correlated. Right panel shows areas (overlain in black) containing no significant local correlation (i.e., mitotic angles are random). The coordinate system used is indicated in the top left-hand corner. The randomness ratio (proportion of black area to total area) calculated at a significance level of 0.01, ranged from 14% to 21% between the fields of view analysed. The raw tracking and mitosis position data for **C–F** can be found in S3 Data. Error bars represent SEM. Scale bar in **A** = 50 μm; scale bar in **B** = 100 μm. A-P, anterior-posterior; SEM, standard error of the mean.

unique to ex vivo culture, we imaged intact E13.5 TCF/Lef::H2B-GFP embryos using 2-photon microscopy (Fig 3B) and tracked mesenchymal cell behaviour for 2 h. Here, we observed the same stereotyped planar mitotic displacement mechanism (Fig 3C and 3D). Global mesenchymal cell division angles were not aligned with the cardinal axes of the embryos (Fig 3E). However, spatial statistical analysis confirmed that contiguous zones of coherent, aligned cell divisions were present (Fig 3F), with areas of correlated mitoses accounting for greater than 80% of the total area in our images. This suggests that the influence on mitosis angle that is global in our 2D skin cultures is instead more regional in intact and growing embryos, perhaps reflecting their more complex topology.

## The dispersal of newly born mesenchymal cells is mediated through a WNT-dependent mechanosensitive mechanism

Having observed correlation of mitotic angles in small domains surrounding the dermal condensates and in larger zones in their absence, we hypothesised that they were likely influenced by mechanical forces. To test this, we compared embryonic mouse skin that had been relaxed fully in culture (S4A Fig) with skin stretched to approximately twice its length along a single axis (S4B Fig). Relaxed skin carried nuclei with a generally oval morphology and no alignment of orientation, while stretched skin showed alignment of the long axis of the mesenchymal nuclei with the applied tension (Figs 4A, S4C, and S4D), though more pronounced for a lateral (dorsal-ventral) stretch than an equivalent axial (anterior-posterior) stretch. A lateral stretch also caused an increase in average nucleus length (Fig 4B). Immunofluorescent detection of Collagen-I confirmed that fibres of the extracellular matrix had aligned with the direction of stretch (S4E and S4F Fig), as had the actin cytoskeleton (S5 Fig). However, we did not observe activation of the mechanotransducer YAP1 [25] in the mesenchyme upon stretching of the skin for 1 h, while an epithelial YAP1 activation response was detected (S5 Fig).

We then tracked cell division and migration in relaxed and stretched E13.5 skin cultures. In relaxed skin, cell migration and mitotic displacement occurred with the same characteristics as observed earlier (Figs 1C, 1D, and S6), but with no preferred angle of mitotic division or migration in dividing cells. Non-dividing cells exhibited a slight preference for anterior-posterior (A-P) movement (Fig 4C). In stretched skin, mitotic displacements and the migration of the newly born and the non-dividing cells aligned with the direction of applied stretch (Fig 4D and 4E and S9 Video). Mitotic displacements in chicken skins also aligned with the direction of tension (S4G Fig). As observed for nuclear elongation, a lateral stretch was more effective in aligning mitotic angles than an equivalent axial stretch (Fig 4F), and relaxed skins showed a similar global alignment score to that from our whole embryo imaging analysis (Figs 3E, 4C, and 4F). We found that suppression of WNT secretion in skins stretched to the same extent, through pharmacological inhibition of the WNT acyltransferase PORCN using LGK-974 (S7 Fig), reduced the alignment of mitoses with the direction of stretch, indicating a role for WNTs in sensing and responding to mechanical cues by mesenchymal cells (Fig 4F). We also identified priming phosphorylation of LRP6 in embryonic mesenchymal cells at mitosis (S8 Fig), previously reported in a range of systems to confer cell cycle based WNT pathway augmentation [26], suggesting that cells in the developing skin are sensitised to WNT signals immediately prior to their division.

Strikingly, we found a unique responsiveness of newly born cells to tissue strain, in which daughter cells move more rapidly in stretched skin compared to relaxed skin, while mother and non-dividing cells did not change their speed of movement. This phenomenon relies on WNT availability, as PORCN inhibition selectively inhibits the post-mitosis migratory behaviour, but not the speed of mother or non-dividing cells (Fig 4G). This reveals that newly born

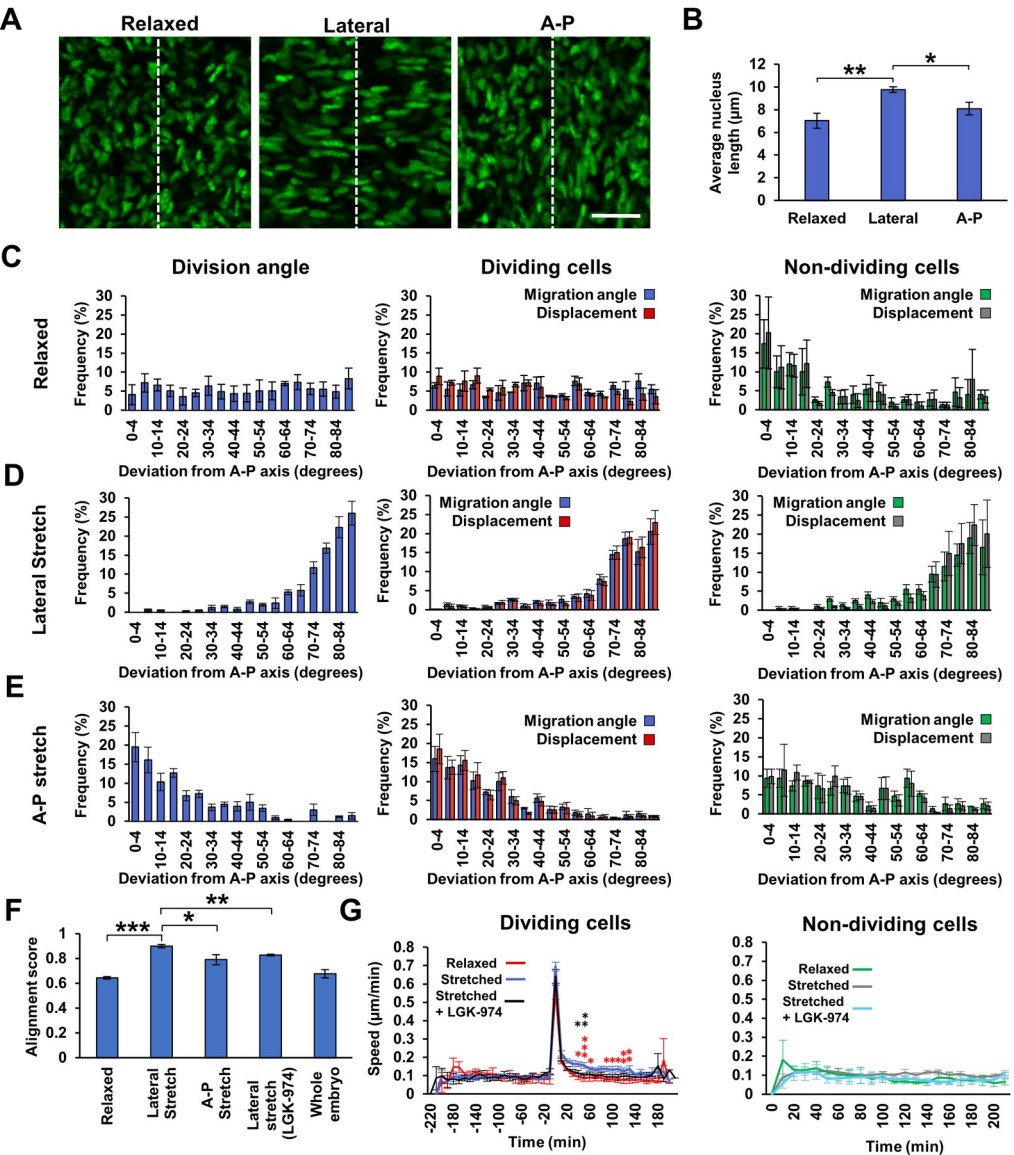

**Fig 4. Orientation of mesenchymal mitosis and newly born cell migration by mechanical cues. (A)** E13.5 TCF/Lef::
H2B-GFP mesenchymal nuclei in skin in relaxed and stretched states. Skins were stretched to approximately twice their
length along a single axis. White dashed line indicates the A-P axis. **(B)** Nucleus length from relaxed ($n$ = 3), laterally
stretched ($n$ = 4), and A-P stretched ($n$ = 3) skins. **(C, E)** Plotted deviations of cell division angles (left) and migration
angle of dividing (middle) and non-dividing (right) cell migration angle from the A-P axis and total cell displacement
for each angle in skins that were **(C)** relaxed ($n$ = 3, average number of dividing cells tracked/video = 67, non-dividing
cells tracked/video = 50), **(D)** stretched laterally, ($n$ = 4, average number of dividing cells tracked/video = 72, non-
dividing cells tracked/video = 50) and **(E)** stretched along the A-P axis ($n$ = 3, average number of dividing cells tracked/
video = 79, non-dividing cells tracked/video = 50). **(F)** Global alignment scores across entire skin of mitosis angles from
relaxed skins ($n$ = 3), laterally stretched skins ($n$ = 4), A-P stretched skins ($n$ = 3), laterally stretched skins from LGK-974
treated embryos ($n$ = 4), and whole embryos ($n$ = 3). **(G)** Speed of tracked dividing (left panel; time 0 = point of mitosis)
and non-dividing (right panel) cells in skins that were relaxed ($n$ = 3, average number of dividing cells tracked/
video = 67), stretched ($n$ = 7, average number of dividing cells tracked/video = 78), or stretched after LGK-974
treatment ($n$ = 4, average number of dividing cells tracked/video = 53); 50 non-dividing cells tracked/video for all
conditions. *$p$ < 0.05, **$p$ < 0.01, ***$p$ < 0.001. The raw numerical values and tracking data for **B–G** can be found in S4
data. Error bars represent SEM. Scale bar in **A** = 20 μm. A-P, anterior-posterior; SEM, standard error of the mean.

mesenchymal cells enter a WNT-dependent mechanosensitive state in early G1 phase, responding to tissue strain with increased migration rates (Fig 4G).

## Discussion

We report a stereotypical migratory behaviour of early G1 phase mesenchymal cells that drives their displacement in tissue growth and patterning. These cells separate rapidly at mitosis and over the next 180 min migrate with greater speed and persistence than their peers further into G1 and S phases. This rapid migratory behaviour is not a continuation of cytokinesis, but instead cells move rapidly and engage in one or more saltatory leaps in the 180 min after mitosis. Cell division and movement is planar, occurring almost entirely in parallel with the epithelium. This stability of mesenchymal cell vertical positions relative to the epithelium allows positional signals, such as epithelial WNT, to direct the appropriate differentiation of distinct layers of the dermis [1,27].

The behaviour of mesenchymal cells at mitosis is highly mechanosensitive, orienting division and migration along lines of tension in the tissue. Mechanosensitivity, or mechanoresponsiveness, requires WNT availability, and WNT signalling likely varies across the cell cycle in developing skin through accumulation of phospho-LRP6 at mitosis to prime WNT reception in early G1 [26]. The contribution of distinct WNT downstream pathways to mesenchymal cell behaviours, and their possible WNT concentration-dependence at different depths of dermis, remains to be elucidated. However, the effect of PORCN inhibition we observe suggests that the focal dermal hypoplasia observed in Goltz syndrome, caused by mutation of *PORCN* (OMIM 305600), may be contributed to by locally dysregulated mesenchymal cell movement.

The behaviour of mesenchymal cells at mitosis defined here matches the long-recognised orientation of dividing cells in epithelial sheets, which align their plane of division perpendicular to orientation of tissue tension [28]. Epithelial cells remain adjacent or close to one another after division, leading to the formation of contiguous epithelial clones indicative of the mode of tissue growth, such as the sharply demarcated lines of Blaschko in the dorsal skin of mammals [20,29]. However, in the mesenchyme, cell division is followed by directed dispersal of newly born daughters, immediately spreading them across the tissue. This mechanism will effectively fill gaps [30] and enforce the disruption of mesenchymal cell clones, preventing formation of mesenchymal condensates through clonal expansion.

Previous work identified an increased frequency of daughter cells from recent cell divisions within mesenchymal condensates [19,21]. This result is explained by our observations as resulting from the increased dispersal of newly born cells compared to their mid and late G1, and S phase peers, making them more likely to encounter recruitment signals. Our simulations show that this observed behaviour alone, without a need to invoke neither spatially heterogeneous proliferation nor a cell cycle phase-specific sensitivity to placode-derived recruitment signals, is sufficient to account for the increased representation of products of recent cell divisions in dermal condensates.

Future work will address the mechanistic basis of this phenomenon and its reliance on WNT availability, its generality in the body's diverse mesenchymal populations, the relationship between intracellular events and extracellular matrix interaction and sensing, and whether orientation of mesenchymal cell division and interphase migration can be modulated by extracellular signals to achieve polarised tissue growth.

## Materials and methods

### Ethics statement

All animal work was conducted under approval of the Animal Welfare and Ethical Review Body (AWERB) at The Roslin Institute, University of Edinburgh, under the project license

(PPL) P682B81E4 provided by the United Kingdom Home Office in accordance with the Animals (Scientific Procedures) Act 1986. Animals underwent euthanasia by cervical dislocation according to Schedule 1 of the Animals (Scientific Procedures) Act 1986.

## Animals

TCF/Lef::H2B-GFP embryos were obtained by crossing male hemizygous TCF/Lef::H2B-GFP mice on a FVB/N genetic background with female FVB/N mice. mTmG mice were on a C57Bl/6JCrl background. Intragastric gavage using LGK-974 (Selleckchem, Houston, Texas, United States) at a dose of 8 mg/kg was performed twice on pregnant mice within a 24 h period before harvesting embryos. LGK-974 was dissolved in 0.5% methylcellulose (Sigma-Aldrich Burlington, Massachusetts, USA), 0.5% Tween-80 (Sigma-Aldrich) in water. Skin collected from LGK-974 treated embryos was maintained in culture medium supplemented with 20 μm LGK-974.

Hyline non-transgenic, transgenic green fluorescent (CAG-GFP), TPZ, Chameleon and membrane GFP reporter chicken eggs were obtained from the Roslin Institute National Avian Research Facility (NARF). Chicken eggs were incubated vertically at 37.8˚C.

## Tissue explant culture and stretching experiments

Dorsolateral skin was dissected from embryos in PBS and attached onto MF-Millipore nitrocellulose filters, 0.45 μm pore size (Catalogue number: HABP04700; Millipore, Burlington, Massachusetts, USA). Explants were cultured in Dulbecco's Modified Eagle Medium (DMEM; Sigma-Aldrich) supplemented with 2% foetal bovine serum (FBS; Thermo Fisher Scientific, Waltham, Massachusetts, USA) and 1% penicillin-streptomycin (Thermo Fisher Scientific) at 37˚C, 5% $CO_2$.

For stretching experiments, skin was dissected and left free-floating for 20 min to permit full relaxation of the tissue. Skin explants were then manually stretched onto a nitrocellulose filter using fine-tipped forceps. For investigation of non-phospho (active) YAP1, skin explants were cultured in either a stretched or relaxed state for 1 h at 37˚C, 5% $CO_2$ prior to fixation.

## mTmG cell labelling

To sparsely label cells with membrane GFP in mTmG mouse skin, explants from embryonic day (E) 13.5 were dissected as previously described, before being gently placed epidermis side down. The dermis was injected with approximately 2 to 5 μl TAT-Cre recombinase (200 U/ml; Sigma-Aldrich) using a glass microcapillary pipette. Explants were then attached to nitrocellulose filters epidermis side up and incubated for 24 h at 37˚C, 5% $CO_2$ before being imaged.

## Chameleon cell labelling

Chameleon chicken line eggs were incubated until E3 before being windowed and the somites injected with approximately 2 to 5 μl TAT-Cre recombinase (200 U/ml) using a glass microcapillary pipette. Eggs were resealed and incubated at 37.8˚C until E7 when skin explants were dissected and attached to nitrocellulose filters as described above.

## Immunofluorescent detection and actin staining

Samples for whole-mount immunofluorescence were fixed overnight in 10% neutral buffered formalin, washed in PBST (PBS + 0.5% Triton ×100), and incubated in blocking buffer 1 (10% goat serum/PBST) for 2 h at room temperature. Samples were incubated in mouse anti-Collagen type I (1:50 dilution; Cat#AB765P; Sigma-Aldrich), rabbit polyclonal anti-phospho-LRP6 (S1490) (1:100 dilution; Cat#2568S, Cell Signaling Technology, Danvers, Massachusetts, USA),

rabbit anti-phospho-histone H3 (S10) (1:100 dilution; Cat#9701, Cell Signaling Technology), mouse anti-non-phosphorylated (active) YAP1 (1:100 dilution; Cat#sc-101199, Santa Cruz) in blocking buffer 1 overnight at 4˚C, washed in PBST and incubated in blocking buffer 1 containing secondary antibodies (goat anti-mouse IgG Alexa Fluor 647 (1:250 dilution; Cat#A21236, Thermo Fisher Scientific), goat anti-rabbit IgG Alexa Fluor 546, (1:500 dilution; Cat#A21245, Thermo Fisher Scientific), goat anti-mouse IgG, Alexa Fluor 488, (1:500 dilution; Cat#A11029, Thermo Fisher Scientific), and Alexa Fluor 647 phalloidin (1:200; Cat#A22287, Thermo Fisher Scientific) overnight at 4˚C. Samples were washed in PBST, counterstained with DAPI (Sigma-Aldrich), washed in PBST, and mounted on Superfrost slides (Thermo Fisher Scientific) with ProLong Gold Antifade mountant (Thermo Fisher Scientific).

Samples for immunofluorescence on sections were fixed overnight in 10% neutral buffered formalin, before being processed into paraffin wax. Antigen retrieval was performed on dewaxed 6 μm sections using an Antigen Retriever 2100 (Aptum Biologics, United Kingdom) and citrate buffer. Sections were then washed in TBS, 0.1% Triton X-100 for 10 min before being washed in TBS, 0.1% Tween 20 (TBST) and incubated in blocking buffer 2 (5% goat serum/TBST) for 1 h at room temperature. Sections were then incubated in anti-mouse non-phosphorylated (active) β-catenin ABC (1:200 dilution; Cat#05–665, Sigma-Aldrich) and rabbit anti-LEF1 (1:200, Cat# ab137872, Abcam) in blocking buffer 2 overnight at 4˚C, before being washed in TBST and incubated in blocking buffer 2 containing secondary antibodies for 1 h at room temperature. Samples were then washed in TBST, counterstained with DAPI, and mounted using ProLong Gold Antifade mountant.

## Imaging

Live time-lapse imaging sequences of TCF/Lef::H2B-GFP mouse and Chameleon chicken skin explants from Glover and colleagues and Ho and colleagues, respectively, [7,12] were used for tracking of dividing and non-dividing cells for Figs 1 and 2. Images were captured at 10 min intervals for mouse time-lapse imaging and at 15 min intervals for chicken.

For time-lapse imaging of mTmG explants and stretched TCF/Lef::H2B-GFP dorsal skin, explants on membrane filters were submerged in DMEM, supplemented with 2% FBS, and 1% penicillin-streptomycin, in glass-bottomed 24-well plates (Eppendorf, Hamburg, Germany), epidermis down. Explants were immobilised using a black plastic washer and imaged at 10 min intervals using an LSM-880 confocal microscope (Zeiss, Oberkochen, Germany) in an incubation chamber at 37˚C.

For whole embryo live time-lapse imaging experiments, embryos were harvested in PBS and affixed to a Petri dish using superglue, before being submerged in DMEM, 2% FBS, and 1% penicillin-streptomycin. Embryos were imaged at 5 min intervals using an LSM 7 MP multiphoton microscope (Zeiss) with a perfusion system to maintain embryo temperature at 37˚C (see Fig 3B).

The percentage decrease in mouse skin explant size after dissection was determined by imaging freshly dissected explants at 3 min intervals using an Axiozoom V16 (Zeiss) and averaging 5 measurements across the A-P axis and perpendicular to the A-P axis for each time point. The degree of stretch was obtained by imaging explants before and after stretching, averaging 5 measurements across the A-P axis and perpendicular to the A-P axis each.

Fixed tissue samples were imaged using an LSM-880 confocal microscope.

## Cell tracking and division analyses

Cell tracking and division analysis was performed using custom written macros for ImageJ. Source code is available through GitHub (https://github.com/richiemort79/mitosis_tools) and

Zenodo (https://doi.org/10.5281/zenodo.8229691). To analyse mitosis angles, maximum intensity Z-projections of time-lapse sequences were drift-corrected, using the "Correct 3D Drift" plugin (https://imagej.net/plugins/correct-3d-drift). The condensate positions (if condensates were present) were recorded and all visible mitoses were mapped by connecting the middle of each daughter nuclei in the first frame after cytokinesis with a line. The angle of this line relative to x and y axes of the imaging frame was calculated using ImageJ.

To track cell behaviour, all visible dividing cells, their resulting daughter cells and at least 50 non-dividing cells were tracked manually until either the video ended, the cell entered a condensate, or the cell migrated beyond the boundary of the video. The mean squared displacement (MSD) of migrating cell populations was calculated using the time ensemble averaging approach [31].

### Spatial analyses

Experimental angular variograms [32] were produced on a point process represented by the mitosis locations in relation to the follicle, i.e., the variogram bins are calculated not between mitoses but between mitoses and follicle. Directional variograms are calculated considering the mitoses at 4 directions in respect to the follicle: 0, 45, 90, and 135 degrees, each of which within a cone of + and − 22.5 degrees. Experimental omnidirectional and directional variograms were estimated using the method of moments and fitted with exponential function by minimization of the weighted (by number of pairs in each bin) sum of squares [33]. From the fitted variograms, the spatial range of influence of the follicle on the variance of the mitoses directions and the explained variance of the mitoses directions by follicle distance has been estimated. The latter is expressed in terms of nugget-to-sill ratio [34].

Permutation analyses were employed to identify whether the modelled variogram (and its parameters) were statistically different, at a significance level of 0.001, from a random process [35]. Directions were permuted across all mitoses, and for each permutation, the experimental variogram was estimated and fitted. This process was repeated 1,000 times. We estimated the Monte Carlo probability that permuted allocations of the directions produced smaller ranges, i.e., the probability that follicles were less influential on the mitoses direction. This is calculated as the number of permutations with lower spatial range divided by the total number of permutations (1,000). If this value is lower than 0.001 then the null hypothesis (no influence) is rejected.

Maps of the direction of the mitoses around the follicle (Fig 2C) were produced by conventional ordinary kriging.

### Local alignment analysis of in vivo embryo mitoses

The random process of embryo mitotic directions, conditional to the mean and variance of the embryo mitoses, was estimated by permuting the directions and predicting them in a 100 by 100 pixel grid via conventional ordinary kriging. The latter uses the modelled variogram of the in vivo embryo mitoses (unchanged). This process was repeated 100 times and the predicted average direction was estimated for each pixel [36]. Finally, we tested if the mean of the distribution of predicted mitosis directions derived from a random process in each pixel was different from the experimental mean of the direction of the mitoses in the same pixel [37]. The map of pixels containing directions that were not significantly different (at a 0.01 significance level) from a random process is shown in Fig 3F.

### Nuclei length and orientation analyses

Per skin sample, nucleus shape and orientation of the longest nucleus axis of 50 individual cells were measured using ImageJ. The length and angle of the cells were gathered and compared to the A-P axis direction.

## Alignment score analysis

Global alignment quantification for cell division angles and orientation of nuclei was conducted by analysing the vector movement of all nuclei within a dataset. Using all paired permutations of vectors and calculating the respective dot products, we computed an overall "alignment coefficient":

$$d = \frac{\sum_{i=1}^{i=N} \frac{\sum_{j=1}^{j=N}}{N}}{N}.$$

Where i and j = the upper and lower bounds, respectively, of the summation series, and $N$ = total number of permutations of vector pairs. An alignment coefficient of 1 represents an ideally linear case, where all vectors are oriented across 1 primary angle. An alignment coefficient of 0 represents ideal isotropy where we observe a truly random distribution of vectors.

We used a colour-based segmentation approach to compute the overall alignment of stained Collagen fibrils. Images were converted from the RGB colour space to the L*a*b colour space where we isolated the *a-channel* and negative region of the *b-channel*. Images were converted to a grayscale representation and passed through a median filter of order $n = 2$ to minimise the levels of salt and pepper noise. Contrast and brightness enhancement was applied with island size removal performed to maximise the isolation of fibrils from the background and nuclei. Images were converted to the frequency domain using a 2D Fourier transformation followed by transformation to polar coordinates. Using the polar form, the column intensities of all pixels were summed, representing the overall image intensity at each 1-degree bin size. The intensity values obtained relate to the proportion of Collagen fibres that are oriented across each specific bin. These values were then used to calculate an overall alignment coefficient.

## Statistical methods

To compare the rate of condensate entry between newly born daughter and non-dividing cells, Fisher's exact tests were performed between the frequency of daughter cells entering a condensate in the first 180 min post division and the frequencies of non-dividing cells entering the condensate in 180 min time windows throughout the imaging period. Fisher's exact tests were also performed to assess timing of condensate entry after mitosis between the frequencies of dividing cells entering a condensate in the first 180 min post division and the frequencies of the same class of cells in later 180 min time windows.

Unpaired Student's *T* tests were performed to determine *p*-values for average nucleus length values and alignment scores.

For comparing cell migration speed in stretched and relaxed skins, log cell speed was analysed using a mixed model with: frame, group, the frame x group interaction and log pre-division speed (based on the mean of frames −4 to −1) fitted as fixed effects; and skin piece, family, cell ID, and the "frame x skin piece" interaction fitted as random effects. A "family" is made up of a parent cell and its 2 daughters. Inclusion of the random effects allowed for additional random variation between skin pieces, family, and cell IDs. In order to avoid the minus infinity values occurring for zero speeds, 0.01 was added to speeds before taking logs. The analysis was carried out using the MIXED procedure in the SAS/STAT software package, Version 9.4.

## Supporting information

**S1 Fig. Newly born daughter cells migrate extensively after division. (A)** Schematic depicting the position of mesenchymal cells migrating and dividing in a planar manner, parallel to

the epidermis. Orientation of view is shown. **(B)** Quantification of cells remaining in-plane after 3 h of imaging (number of tracked daughter cells = 100; z-slice depth = 6 μm). **(C)** Mitosis plots mapping mother and daughter tracks before and after cell division, with corresponding cell speed plots over the time course of imaging below. Tracks were chosen from cells in which cell division took place within a time point of 40%–60% through the imaging, and both daughter cells were tracked until the end of the video. The raw numerical values and tracking data for **B** and **C** can be found in S1 Data.
(TIF)

**S2 Fig. Newly born mesenchymal cells exhibit rapid migration in embryonic chicken skin.** **(A)** Single z-planes from confocal time-lapse series of an E6 Chameleon chicken skin explant culture treated with cell permeant TAT-Cre and cultured for 24 h. Magenta dots highlight mother and daughter cells. **(B)** Diffusion plots showing the migration direction and dispersion of tracked mother (left; $n = 62$), daughter (middle; $n = 124$), and non-dividing (right; $n = 50$) cells from a single skin explant. **(C, D)** Speed **(C)** and Euclidean distance travelled **(D)** of dividing (left panels; time 0 = point of mitosis) and non-dividing (right panels) cells from Chameleon chicken skin explants. **(E)** Persistence (Euclidean/accumulated distance) of mother, daughter, and non-dividing cells against time. **(F)** Mean squared displacement (MSD) of mother cells (Mo; $n = 62$) for 180 min prior to division, daughter cells for 180 min after division, daughter cells for 180–360 min after division (Da; $n = 124$ in both windows), and non-dividing cells (ND; $n = 50$) for a single representative skin explant. **(G)** Mean diffusion coefficient (the slope of the line in **F**) of mother cells (Mo), daughter cells (Da) for 180 min after division, daughter cells for 180–360 min after division, and non-dividing cells (ND). A one-way analysis of variance (ANOVA ($p < 0.01$)) followed by pairwise post hoc Tukey's honestly significant difference tests revealed a significant difference between daughters (0–180 min) and mother cells, and between daughters (0–180 min) and non-dividing cells (*$p < 0.05$, **$p < 0.01$). The raw tracking data for **B**–**G** can be found in S5 Data. **(H)** Confocal images of clonally labelled mesenchymal (left) and epithelial (right) cells in E7 Chameleon skin explant culture following somite and overlying epithelium injection of cell permeant TAT-Cre at E3. **(I)** Tissue derivatives of a tdTomato somite (TPZ transgenic line) transplanted into a CAG-GFP host embryo, at E6.5. Left panel, intact embryo; right panel, isolated skin. **C, D, E,** and **G** time-lapse videos $n = 3$ from 3 independent samples, mean number of dividing cells tracked per video = 60, mean number of non-dividing cells tracked per video = 91. Error bars represent SEM. Scale bar in **A** = 20 μm, scale bar in **H** = 200 μm, scale bar in **I** = 2 mm.
(TIF)

**S3 Fig. Collagen-I is reduced in dermal condensates.** Single planes and orthoviews from confocal imaging of Collagen-I immunofluorescence in E14.5 TCF/Lef::H2B-GFP skin explant stained with phalloidin (to detect F-actin) and DAPI. White dashed lines indicate the plane of the z-section. Scale bar = 50 μm.
(TIF)

**S4 Fig. Mesenchymal nuclei and Collagen-I align with mechanical strain. (A)** Plot showing the percentage decrease in anterior-posterior (A-P) and dorsal-ventral (Lat) length of mouse skin explants in a 30 min period ($n = 5$) when suspended freely in culture medium. **(B)** Images of E13.5 TCF/Lef::H2B-GFP mouse skin explants before and after a lateral stretch (upper panels) or a stretch along the anterior-posterior (A-P) axis of the embryo (lower panels). White dashed arrows show direction of stretch. **(C)** Nucleus orientation angle for relaxed ($n = 3$; upper), lateral stretched ($n = 4$; middle), and A-P stretched ($n = 3$; lower) skins. **(D)** Alignment score of nucleus orientation angle from relaxed ($n = 3$), laterally stretched ($n = 4$), and A-P

stretched (*n* = 3) skins. **(E)** Single planes from confocal imaging of Collagen-I immunofluorescence in E13.5 TCF/Lef::H2B-GFP skin explants in stretched states. Dashed white line indicates direction of stretch. **(F)** Alignment scores of Collagen fibres from skin samples shown in **E.** The raw numerical values for **A**, **C**, **D**, and **F** can be found in S4 Data. **(G)** Single planes from confocal imaging of an A-P stretched E7 membrane GFP (mGFP) chicken skin. Daughter nucleus pairs are connected by white lines, showing coherent angles of mitosis aligned with applied tension. Error bars represent SEM. Scale bar in **B** = 2 mm; scale bar in **E** = 100 μm; scale bar in **G** = 50 μm.
(TIF)

**S5 Fig. YAP1 is activated in the basal epidermis but not in mesenchymal cells after 1 h of skin stretching.** Single planes of the dermis and basal epidermis from confocal imaging of active (non-phospho) YAP1 immunofluorescence in E13.5 mouse skin explants, stained with phalloidin (to detect F-actin) and DAPI, in relaxed and laterally stretched states. Scale bar = 50 μm.
(TIF)

**S6 Fig. Newly born cells in stretched skin show increased speed and displacement. (A**, **B)** Speed **(A)** and Euclidean distance travelled **(B)** of tracked dividing (top panels; time 0 = point of mitosis) and non-dividing (bottom panels) cells in skins that were relaxed (*n* = 3, average number of dividing cells tracked/video = 67, non-dividing cells tracked/video = 50), stretched laterally, (*n* = 4, average number of dividing cells tracked/video = 72, non-dividing cells tracked/video = 50), and stretched along the A-P axis (*n* = 3, average number of dividing cells tracked/video = 79, non-dividing cells tracked/video = 50). The raw tracking data for **A** and **B** can be found in S4 Data. Error bars represent SEM.
(TIF)

**S7 Fig. Suppression of active β-catenin in embryos of LGK-974 treated pregnant mice.** Confocal imaging of active β-catenin (ABC) and LEF1 immunofluorescence in E13.5 TCF/Lef::H2B-GFP skin sections, stained with DAPI, from embryos from pregnant mice untreated (control) or treated with LGK-974 for 24 h. Less active β-catenin signal is detected in the mesenchyme of treated embryos. Scale bar = 50 μm.
(TIF)

**S8 Fig. Detection of phospho-LRP6 (serine 1490) in mesenchymal and basal epidermal cells at mitosis. (A**, **B)** Single planes of the dermis and basal epidermis from confocal imaging of phospho-LRP6 (S1490) **(A)** and phospho-histone H3 (S10) **(B)** immunofluorescence in E13.5 mouse skin explants, stained with phalloidin (detecting F-actin) and DAPI. Phospho-histone H3 signal illustrates the morphology of nuclei undergoing mitosis. Yellow arrows indicate cells that are in mitosis and have condensed chromatin. Phospho-LRP6 (S1490) is detected at highest levels in cells undergoing division in mesenchyme and epithelium. Scale bar = 20 μm.
(TIF)

**S1 Table. Statistical comparisons of rate of cell entry to dermal condensates in 180 min windows for dividing and non-dividing cells.** Newly born daughter cells have a higher rate of condensate entry compared to non-dividing cells in their first 180 min.
(XLSX)

**S1 Video. Time-lapse imaging of dividing cells in TCF/Lef::H2B-GFP mouse skin.** Confocal time-lapse imaging of a cultured E13.5 TCF/Lef::H2B-GFP skin explant. Nuclei in different z-planes are colour-coded (red = closest to epithelium, green = middle, grey = plane furthest

from epithelium). Magenta, cyan, and red dots highlight nuclei undergoing mitosis; magenta, cyan, and red circles indicate the point at which cells divide. In the second round of playing dividing cell migration tracks are in blue and non-dividing cell tracks in green.
(AVI)

**S2 Video. Fast-moving cell in dermis.** Confocal time-lapse imaging of a cultured E13.5 TCF/Lef::H2B-GFP skin explant. Nuclei in different z-planes are colour-coded (red = closest to epithelium, green = middle, grey = plane furthest from epithelium). A fast-moving cell is highlighted by a magenta dot and circle.
(AVI)

**S3 Video. Daughter cells pause after cytokinesis before making large migration movements.** Confocal time-lapse imaging of a cultured E13.5 TCF/Lef::H2B-GFP skin explant. Nuclei in different z-planes are colour-coded (red = closest to epithelium, green = middle, grey = plane furthest from epithelium). Magenta and cyan dots highlight dividing cells that pause after division before moving rapidly. Magenta and cyan circles indicate the point at which cells divide. In the second round of playing dividing cell migration tracks are in blue and non-dividing cell tracks in green.
(AVI)

**S4 Video. Time-lapse imaging of dividing cells in chicken skin.** Confocal time-lapse imaging of sparsely labelled and cultured E6.5 Chameleon chicken skin explant. Magenta, cyan, and red dots highlight cells undergoing mitosis; magenta, cyan, and red circles indicate the point at which cells divide. Note that the entire cell volume is labelled with these cytoplasmic fluorescent proteins.
(AVI)

**S5 Video. Time-lapse imaging of dividing mesenchymal cells in mTmG mouse skin.** Confocal time-lapse imaging of cultured E13.5 mTmG skin explant. Magenta and cyan dots highlight mesenchymal cells undergoing mitosis, magenta and cyan circles indicate the point at which cells divide. White arrowhead points to a cell process that persists through mitosis.
(AVI)

**S6 Video. Time-lapse imaging of dividing epithelial cell in mTmG mouse skin.** Confocal time-lapse imaging of cultured E13.5 mTmG skin explant, showing an example of a peridermal cell undergoing mitosis. The daughter cells remain in contact with one another.
(AVI)

**S7 Video. Time-lapse imaging of cells dividing within a dermal condensate in TCF/Lef::H2B-GFP mouse skin.** Confocal time-lapse imaging of a cultured E13.5 TCF/Lef::H2B-GFP skin explant. Magenta and cyan dots highlight nuclei that have divided within a condensate, magenta and cyan circles indicate the point at which cells divide. In contrast to the general dermal mesenchyme, dividing cells within dermal condensates produce daughters that remain in close proximity to one another.
(AVI)

**S8 Video. Time-lapse imaging of peripheral cells dividing tangentially around a dermal condensate in TCF/Lef::H2B-GFP mouse skin.** Confocal time-lapse imaging of a cultured E13.5 TCF/Lef::H2B-GFP skin explant. Nuclei in different z-planes are colour-coded (red = closest to epithelium, green = middle, grey = plane furthest from epithelium). Magenta and cyan dots highlight nuclei undergoing mitosis around a condensate, magenta and red circles indicate the point at which cells divide. Cells that are peripheral to the condensate have

elongated nuclei running at a tangent to the condensate and divide along this axis. (AVI)

**S9 Video. Time-lapse imaging of cell divisions aligning with stretch in TCF/Lef::H2B-GFP mouse skin.** Confocal time-lapse imaging of a cultured E13.5 TCF/Lef::H2B-GFP skin explant stretched along the dorsolateral axis. Magenta circles highlight nuclei undergoing mitosis, in which the division angle aligns with the direction of applied stretch. (AVI)

**S1 Appendix. Description of agent-based mathematical model used to generate data for Fig 2D–2H.** (PDF)

**S1 Data. Data pertaining to Figs 1B–1G, 2A, 2F, 2H, S1B and S1C.** (XLSX)

**S2 Data. Data pertaining to Figs 2B, 2C, and 3A.** (XLSX)

**S3 Data. Data pertaining to Fig 3C–3F.** (XLSX)

**S4 Data. Data pertaining to Figs 4B–4, S4A, S4C, S4D, S4F, S6A, and S6B.** (XLSX)

**S5 Data. Data pertaining to S2B–S2G Fig.** (XLSX)

## Acknowledgments

We thank staff of the Biological Research Facility and National Avian Research Facility at the Roslin Institute for technical support and Lorraine Rose for providing mTmG mouse embryos. We thank Raphael Voituriez for insightful discussion on cell movement and James Headon for help with image processing. This research made use of the Balena High Performance Computing (HPC) Service at the University of Bath.

## Author Contributions

**Conceptualization:** Jon Riddell, Christian A. Yates, Richard L. Mort, Denis J. Headon.

**Data curation:** Jon Riddell, Shahzeb Raja Noureen, Luigi Sedda, Richard L. Mort, Denis J. Headon.

**Formal analysis:** Jon Riddell, Shahzeb Raja Noureen, Luigi Sedda, Connor A. Bain, Arianna Berbeglia, Helen Brown, Calum Anderson, Yuhang Chen, Michael L. Crichton, Christian A. Yates, Richard L. Mort, Denis J. Headon.

**Funding acquisition:** Richard L. Mort, Denis J. Headon.

**Investigation:** Jon Riddell, Shahzeb Raja Noureen, Luigi Sedda, James D. Glover, William K. W. Ho, Connor A. Bain, Arianna Berbeglia, Michael L. Crichton, Christian A. Yates, Richard L. Mort, Denis J. Headon.

**Methodology:** Jon Riddell, Shahzeb Raja Noureen, Luigi Sedda, James D. Glover, William K. W. Ho, Connor A. Bain, Helen Brown, Calum Anderson, Yuhang Chen, Michael L. Crichton, Christian A. Yates, Richard L. Mort, Denis J. Headon.

**Project administration:** Jon Riddell, Denis J. Headon.

**Resources:** Shahzeb Raja Noureen, Luigi Sedda, James D. Glover, William K. W. Ho, Michael L. Crichton, Christian A. Yates, Richard L. Mort, Denis J. Headon.

**Software:** Jon Riddell, Shahzeb Raja Noureen, Luigi Sedda, Christian A. Yates, Richard L. Mort, Denis J. Headon.

**Supervision:** Jon Riddell, Yuhang Chen, Michael L. Crichton, Christian A. Yates, Richard L. Mort, Denis J. Headon.

**Validation:** Jon Riddell, Shahzeb Raja Noureen, Richard L. Mort, Denis J. Headon.

**Visualization:** Jon Riddell, Shahzeb Raja Noureen, Luigi Sedda, Connor A. Bain, Michael L. Crichton, Christian A. Yates, Richard L. Mort, Denis J. Headon.

**Writing – original draft:** Jon Riddell, Shahzeb Raja Noureen, Luigi Sedda, Christian A. Yates, Richard L. Mort, Denis J. Headon.

**Writing – review & editing:** Jon Riddell, Shahzeb Raja Noureen, Luigi Sedda, James D. Glover, William K. W. Ho, Connor A. Bain, Arianna Berbeglia, Helen Brown, Calum Anderson, Yuhang Chen, Michael L. Crichton, Christian A. Yates, Richard L. Mort, Denis J. Headon.

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
