## [Editor Report · Decision Letter 0]

13 Feb 2023

Dear Dr Headon, 

Thank you for submitting your manuscript entitled "Newly born mesenchymal cells disperse through a rapid mechanosensitive migration" for consideration as a Short Report by PLOS Biology. Please accept my apologies for the delay in getting back to you as we consulted with an academic editor about your submission. 

Your manuscript has now been evaluated by the PLOS Biology editorial staff, as well as by an academic editor with relevant expertise, and I am writing to let you know that we would like to send your submission out for external peer review.

Once your full submission is complete, your paper will undergo a series of checks in preparation for peer review. After your manuscript has passed the checks it will be sent out for review. To provide the metadata for your submission, please Login to Editorial Manager (https://www.editorialmanager.com/pbiology) within two working days, i.e. by Feb 15 2023 11:59PM.

Kind regards,

Richard

Richard Hodge, PhD

Associate Editor, PLOS Biology

rhodge@plos.org

PLOS

---

## [Decision Letter · Decision Letter 1]

23 Mar 2023

Dear Dr Headon,

Thank you for your patience while your manuscript "Newly born mesenchymal cells disperse through a rapid mechanosensitive migration" was peer-reviewed at PLOS Biology. Please accept my sincere apologies for the delays that you have experienced during the peer review process. Your manuscript has now been evaluated by the PLOS Biology editors, an Academic Editor with relevant expertise, and by three independent reviewers. 

In light of the reviews, which you will find at the end of this email, we would like to invite you to revise the work to address the reviewers' reports.

As you will see, the reviewers agree that your findings are interesting and well done, but Reviewer’s #2 and #3 note that the study lacks further molecular insights into the mechanism underlying the enhanced migration after cell division. Whilst we appreciate that your manuscript is being considered as a Short Report, after discussions with the Academic Editor, we agree with the reviewers suggestions here to strengthen the molecular characterization and provide some preliminary insights. This would include a molecular characterization to provide more insights into the potential mechanism of the rapid migration after mitosis, and how this then leads to the formation of dermal condensates.

Given the extent of revision needed, we cannot make a decision about publication until we have seen the revised manuscript and your response to the reviewers' comments. Your revised manuscript is likely to be sent for further evaluation by all or a subset of the reviewers.

**IMPORTANT - SUBMITTING YOUR REVISION**

*Re-submission Checklist*

*Published Peer Review*

*PLOS Data Policy*

*Blot and Gel Data Policy*

Sincerely,

Richard

Richard Hodge, PhD

Associate Editor, PLOS Biology

rhodge@plos.org

REVIEWS:

Reviewer #1: In this manuscript, Riddell et al. use live imaging to assess the migratory behavior of mesenchymal cells in mouse and chick skin. They find that mesenchymal cells undergo much faster dispersal in early interphase, within first three hours after cytokinesis, in a sub-phase of G1. The authors posit that this migratory behavior allows the recently divided cells to travel longer distances thereby facilitating their entry to dermal condensates.

This is potentially highly interesting paper describing a novel phenomenon and is by large of high technical quality. The stretching experiments are innovative, although their relevance to formation of dermal condensates could be better clarified. In its current form, there's quite a few points that the authors should address and clarify.

Specific points:

1. While the phenomenon described (fibroblasts moving faster in early G1) by the authors is intriguing, I still wonder how generalizable these findings are, even in the mouse? Imaging of adult mice is unlikely not feasible to the authors, but what about other developmental stages than E13.5 � 14.5?

2. In many cases it is not clear on how many independent experiments the data are based on. Please, include that information in the Figure legends.

3. The authors make a point about mesenchymal cells moving (and dividing) parallel to the epithelium at all depths of mesenchyme, and discuss this in the context of epithelial signaling and differentiation of distinct dermal layers. Maybe I missed something, but I could not find quantifications to back up this statement.

4. Figure 2A and S1 Table reports important data about the likelihood of dividing vs. non-dividing cells to enter the dermal condensate. How far from the placode/dermal condensate were the cells that were included in the analysis shown here? 

5. The previous point leads to my next questions. I am quite confused about data shown in Fig. 2B and C. Based on Fig. 2B, are the authors concluding that the distance from the center on one DC to the center of the next one is on average 600µm? To me this distance sounds terribly long. It seems that division angle from cells as far as 1 mm away from the center of the dermal condensate has been quantified. But wouldn't these be cells that are influenced by other dermal condensates? In Fig. 2C, are the authors stating that within an area of approx. 0.7 x 0.5 mm, one can typically observes only one dermal condensate forming at E13.5 � E14.5? Could the authors explain better the lower part of 2C, please. Why are the angles on the "north-south" axis around 20 degrees and in the "west-east" axis around 70 degrees when the text reads "We also observed that mitoses occur at a tangent to the condensate perimeter". Wouldn't this mean that the angles should be close to 90 degrees everywhere in the vicinity of the DC?

6. The authors claim that collagen-I forms "rings" around the hair bud and that Tcf/Lef1-H2B-GFP+ cells align their nuclei around the placode. However, I have hard times in seeing the collagen "ring" from Fig. 2D. Moreover, H2B-GFP+ nuclei appear 'merged' and blurry in the figure. A higher quality image with orthoviews should be shown. Now it is not clear at what depth is this optical section from (presumable close to the interfollicular epithelium rather than the actual dermal condensate at the bottom of the E15.5 hair bud?). Co-staining with a dermal condensate marker such as Sox2 plus an arrow indicating A-P orientation would be helpful, as E15.5, placodes and DCs are already highly polarized.

7. The authors state that "Global mesenchymal cell division angles were not aligned with the cardinal axes of the embryos (Fig. 3E)". However, it is unclear what is 0 degrees stands for in Fig. 3E.

8. In Discussion, the authors make the following conclusion: "…in the mesenchymal condensate, where the extracellular matrix is depleted." I find this a strong overstatement based on mere absence of collagen-I staining. I guess there could be also other ECM molecules than collagen-I, and I believe the transcriptomic profiling of DC cells is also indicative DC-specific ECM molecules. 

9. Some technical details are missing:

Materials and methods report how whole embryos were immobilized for imaging. But what about explants? It is said that explants on membrane filters were submerged in DMEM in glass-bottomed plates epidermis down. Were the explants immobilized somehow?

Drift correction was made to videos, but it remained unclear how this was made.

Minor comments:

1. Movies like S3 would be easier to follow if they were running twice, the second round including tracks color coded separately for dividing and non-dividing cells.

2. Please orient all images in Fig. 4A the same way with respect to A-P axis as that would make it much easier to appreciate the nuclear orientations.

3. There are some mistakes when referring to the Figures. E.g. in Materials and Methods, it reads "Maps of the direction of the mitoses around the follicle (Fig 2D) were produced by conventional ordinary kriging."

Reviewer #2: Riddel et al., Critique 2023 Plos Biology

The manuscript, authored by Riddell et al., is a characterization of dermal fibroblasts progenitors that contribute to the dermal condensate in mouse and chicken embryonic skin. Their observations show a rapid, persistent, migration in a three-hour period. They found that (a) the newly born dermal mesenchymal cells migrate rapidly and persistently resulting in greater displacement upon and following mitosis compared to the non-dividing cells in mouse embryos. (b) The newly born mesenchymal cells are more likely to encounter the dermal condensates. (c) The orientations of mitoses are aligned in the mouse embryonic mesenchyme cultured ex vivo while the global mesenchymal cell division angles are not aligned in the intact E13.5 embryos, suggesting the mitotic orientation is possibly influenced by the topology of tissue. (d) The orientation of the newly born mesenchymal cells are sensitive to the orientation of tension.

Their findings are interesting, relevant, and provocative. They have elegant imaging techniques to visualize the cell behaviors in the dermis. The major concern with this report is the study design is largely descriptive in wildtype over a short time period predominantly and one aspect such as tension is manipulated in figure4. The authors do not demonstrate which of the variables they have identified is causative for the cell behavior (orientation, collagen pattern, condensate, movement). For instance can they compare a skin model that has condensates versus not in chick or mouse? Or can they use pharmacological inhibition of collagen assembly? Or can they manipulate a specific aspect of mechanosensation (stretch is complex because it can also affect planar cell polarity) ? One of the main findings such as “recruitment” of daughter cells into the condensates concept is not clearly demonstrated in their results or experimental strategies. Couldn’t the spreading migration makes just make it more plausible and convenient for daughter cells to encounter the condensate? 

Here are some additional concerns and suggestions: 

Introduction text: Introduce some more relevant detail and context for the early events in pre-dermal condensate morphogenesis at the cell biology level. Recent review from Myung P. et al., in Development 2022 on this issue might be cited. Why mention dermal adipocyte origin because it is not relevant to this study. What does it mean that there is “high developmental potential” of fibroblasts? 

For Figure 1: 

Figure 1K: Microenvironment analysis is cursory. There can be more differences than just collagen. It does not add to the analysis of the wild-type scenario. There is not that much mature collagen protein. Which type of collagen? In this window there is not that much collagen protein. 

Figure1 will be easier to understand if it could include a simple schematic depicting the position of the ex vivo mouse skin tissue and the mitotic dermal cells they are testing and the displacement direction of those mesenchymal cells. In Fig 1C, it turns out that the migration speed of the non-dividing cells is close to 0, however the Fig 1D indicates that the non-dividing cells still showed increasing Euclidean distance during the time period. And in text, they indicate that the daughter cells result in ‘a mean Euclidean displacement ~50% greater than’ the non-dividing cells, but it is not displayed in Fig 1G.

For Fig 1J and S7 video, authors suggested they observed some mitoses occurring in dermal condensates while cells in dermal condensates are largely quiescent. What is the incidence of mitosis in dermal condensates under their observation? Have they also observed similar mitosis in chicken embryonic skin mesenchymal cells?

Figure 2: 

In text, they suggested that they found the orientation of mitosis is at a tangent to the condensate perimeter and the collagen ring is generated surrounding the condensate, thus they concluded that the extracellular environment surrounding the dermal condensate influences the orientation of cell migration and mitosis. This is not shown with causative data or ruling out other possibilities. But what if the orientations of the cell movement and the mitosis are already set up in a chemotaxis manner for entering and constituting the dermal condensate later? To rule out this possibility, it would be convincing to show the expression of signals such as FGF20 and SHH, or show the cell polarity distribution a day earlier.

Figure 2 text: Does the original orientation of division relative to condensate influences the direction of displacement of the daughter cells? If these cells are not oriented directly towards the condensate (lines 173-176) to begin with, how do they get displaced towards the condensate? Figure 2D: If dermal condensates influence division angle within this radius, can they show this in a causative manner in a system that lacks dermal condensates in chick or mouse. 

Figure 2D: Similarly, if collagen ring surrounding the condensate “influences the orientation of the cells and direction of mitosis” can they show this in a collagen mutant or Loxl2-inhibition models. Which collagen (collagen 1 or 3)? Also, the image is not clearly annotated for peri-condensate nuclei alignment. 

Figure 3: 

In figure 3, the in vivo imaging shows regional alignment of mitotic cells. Where is this in relation to the condensate? If it is regionally aligned, how large are the regions. 

Figure 2C & 3F: Adding the coordinate system that the angles come from would make it slightly quicker to interpret. Like where is 0 degrees?

Figure 4: 

For Fig 4G, how to explain that the non-dividing cells in relaxed tissue show a higher speed than them in stretched tissue in the first 20 mins?

Lines 289 and 290: What makes newly born mesenchymal cells enter a more mechanosensitive state in early G1? It is interesting that the mother cells do not demonstrate the same speed of movement but could they contribute to the mechanism at all of what is going on during G1 to make newly born cells more sensitive?

Reviewer #3: In this report, Riddell et al., described live imaging observation and quantitative analysis of dermal fibroblast cells during dermal condensate formation in embryonic skin. They made an interesting finding that newly divided fibroblast cells tend to migrate more rapidly and have a higher probability to enter dermal condensates than their non-dividing counterparts. The imaging results are of high quality and obtained from multiple systems, including ex vivo cultured mouse skin, live mouse embryos and explant chicken skin culture. The quantitative analysis is well done, and the results are informative. However, this is a largely descriptive study, which can benefit from some molecular characterization.

Major points: 

1. What is the molecular basis for the increased migration immediately after the cell division? A key observation is that newly divided fibroblast cells migrate more rapidly in the first 180min after the division and gradually reduce their motility (Fig. 1C-D). They provided evidence that the lack of Collagen I in dermal condensates, where cell division doesn't lead to rapid migration, is correlated with immobility of fibroblasts within the DC. However, this doesn't explain (at least provide some clues) why fibroblasts outside of the DC gain mobility upon division. Collagen I expression doesn't appear to distinguish dividing vs non-dividing cells. Could it be other cell adhesion molecules or cytoskeleton machinery? 

2. Fig. S3 show the dermal condensate formation over time. However, the display was not very informative, and I can't quite distinguish what is the difference from 0 to 4 hours. They should use 3D visualization to show the dynamics of the DC formation. Based on their description that "DC form from the epithelium downwards", does it suggest newly added cells, some of them should be newly born cells and some of them should be non-dividing cells, are added to the middle to low portion of the DC and only on the periphery? Does the destination vary for newly born vs non-dividing cells? 

3. Single-cell analysis and morphological studies suggest the DC has the highest WNT signaling, which should be reported by their TCF/Lef::H2B-GFP reporter. Do the newly incorporated cells into the DC have higher H2BGFP levels than the ones left behind? 

4. In Fig 4H-I, why do dividing cells have lower probability to enter the DC if they're located very close to the DC? 

5. They further reported that newly born fibroblasts migrate even faster under a stretched condition in Fig. 4G, and they suggested that these cells are at a more mechanosensitive state. To probe deeper, they should examine several markers that are associated with mechanosensitivity of the cells, such as cell adhesion (reduced adhesion?), actin cytoskeleton and YAP signaling pathway, which is known to drive the cell cycle and sensing cell mechanics.

---

## [Decision Letter · Decision Letter 2]

2 Aug 2023

Dear Dr Headon,

Thank you for your patience while we considered your revised manuscript "Newly born mesenchymal cells disperse through a rapid mechanosensitive migration" for publication as a Short Report at PLOS Biology. This revised version of your manuscript has been evaluated by the PLOS Biology editors, the Academic Editor and the original reviewers.

Based on the reviews, I am pleased to say that we are likely to accept this manuscript for publication, provided you satisfactorily address the remaining points raised by the reviewers. Please also make sure to address the following data and other policy-related requests that I have provided below (A-F):

(A) We would like to suggest the following modification to the title: 

“Rapid mechanosensitive migration and dispersal of newly divided mesenchymal cells aid their recruitment into dermal condensates”

(B) In the ethics statement in the Methods section, we would be grateful if you could clarify whether the Animal Welfare and Ethical Review Body is an Institutional Animal Care and Use Committee (IACUC) that reviewed and approved the study. Please also include the specific approval number issued by your ethics committee and the method of euthanasia used to sacrifice the animals. 

(C) You may be aware of the PLOS Data Policy, which requires that all data be made available without restriction: http://journals.plos.org/plosbiology/s/data-availability. For more information, please also see this editorial: http://dx.doi.org/10.1371/journal.pbio.1001797

-Supplementary files (e.g., excel). Please ensure that all data files are uploaded as 'Supporting Information' and are invariably referred to (in the manuscript, figure legends, and the Description field when uploading your files) using the following format verbatim: S1 Data, S2 Data, etc. Multiple panels of a single or even several figures can be included as multiple sheets in one excel file that is saved using exactly the following convention: S1_Data.xlsx (using an underscore).

-Deposition in a publicly available repository. Please also provide the accession code or a reviewer link so that we may view your data before publication. 

Figure 1B-G, 2A-C, 2E-H, 3A, 3C-F, 4B-G, S1B-C, S2B-G, S4A, S4C-D, S4F, S6A-B

(D) Thank you for already depositing the source code used for the cell tracking analyses in Github (https://github.com/richiemort79/mitosis_tools). We ask that you please attach this deposition to the Zenodo repository or similar so that the deposition is issued a DOI and has long term maintenance. 

(E) Please also ensure that each of the relevant figure legends in your manuscript include information on *WHERE THE UNDERLYING DATA CAN BE FOUND*, and ensure your supplemental data file/s has a legend.

(F) Please ensure that your Data Statement in the submission system accurately describes where your data can be found and is in final format, as it will be published as written there. 

We expect to receive your revised manuscript within two weeks. 

*Published Peer Review History*

*Press*

Kind regards,

Richard

Richard Hodge, PhD

rhodge@plos.org

Reviewer remarks:

Reviewer #1: The authors of Riddell et al. have been largely able to respond to the criticism I raised, either by deleting data, providing new data and/or new analysis/images. In S3, I would appreciate seeing the plane of the z-section in the orthoviews to still better appreciate at what depth the optical section is from.

The new data on the role of the Wnt pathway is intriguing, having seen the effect on the migration of newly born cells (as in Fig. 1) without strecth would have interesting.

Reviewer #2: Thank you for the extensive revisions on the text and figures and the inclusion of some functional experiments. The focus on the Wnt pathway is a good addition to the paper and demonstrates the pre-DC events. Your findings are consistent with the scRNA-seq and functional analysis from the Myung and Rendl labs. I apologize for the delay. I did not get the reviewer responses with the manuscript files. 

Reviewer #3: In the revised manuscript, Riddell et al. have strengthened this interesting study. Notably, they have shown that the pharmacological inhibition of Wnt by LGK-974 disrupts the alignment of mitosis along the stretch and reduces the migratory speed of the newly born daughter cells. Furthermore, they have demonstrated that LRP6 phosphorylation peaks at mitosis, providing a plausible mechanism for enhanced Wnt signaling at mitosis, which subsequently affects daughter cell migration. Therefore, I have no further concerns about the manuscript. However, the Wnt-dependent behaviors of cell division and post-division migration are highly interesting. It is known that basal epithelial cells are the major source of WNT ligand production during embryonic development, and this generates a gradient of WNT signaling from the epidermis to the dermis. The authors should discuss whether they observed a differential migratory speed in newly born mesenchymal daughter cells, depending on the distance of these cells from the basal epidermis, and whether WNT high mesenchymal cells (marked by TCF:LEF-H2bGFP reporter) migrate more rapidly in general.

---

## [Editor Report · Decision Letter 3]

29 Aug 2023

Dear Dr Headon,

Thank you for the submission of your revised Short Report "Rapid mechanosensitive migration and dispersal of newly divided mesenchymal cells aid their recruitment into dermal condensates" for publication in PLOS Biology. On behalf of my colleagues and the Academic Editor, Marianne Bronner, I am pleased to say that we can accept your manuscript for publication, provided you address any remaining formatting and reporting issues. These will be detailed in an email you should receive within 2-3 business days from our colleagues in the journal operations team; no action is required from you until then. Please note that we will not be able to formally accept your manuscript and schedule it for publication until you have completed any requested changes.

PRESS

Best wishes, 

Richard

Richard Hodge, PhD

rhodge@plos.org

PLOS
